# Leucocyte Telomere Length and Glucose Tolerance Status in Mixed-Ancestry South Africans

**DOI:** 10.3390/cells8050464

**Published:** 2019-05-16

**Authors:** Cecil J. Weale, Glenda M. Davison, Gloudina M. Hon, Andre P. Kengne, Rajiv T. Erasmus, Tandi E. Matsha

**Affiliations:** 1SAMRC/CPUT/Cardiometabolic Health Research unit, Department of Biomedical sciences, Faculty of Health and Wellness Sciences, Cape Peninsula University of Technology, P.O. Box 1906, Bellville 7530, South Africa; cecilweale@gmail.com (C.J.W.); HonG@cput.ac.za (G.M.H.); 2Division of Haematology, Department of Pathology, Faculty of Health Sciences, University of Cape Town, Cape Town 7925, South Africa; 3Non Communicable Diseases Research Unit, South African Medical Research Council, Cape Town 7505, South Africa; Andre.Kengne@mrc.ac.za; 4Department of Medicine, University of Cape Town, Cape Town 7925, South Africa; 5Department of Pathology, Faculty of Medicine and Health Sciences, National Health Laboratory Service (NHLS) and Stellenbosch University, Cape Town 7505, South Africa; rte@sun.ac.za

**Keywords:** leucocyte telomere length, hyperglycemia, type II diabetes

## Abstract

Telomeres are DNA-tandem repeats situated at the ends of chromosomes and are responsible for genome stabilization. They are eroded by increased cell division, age and oxidative stress with shortened leucocyte telomeres (LTL) being associated with inflammatory disorders, including Type II diabetes. We assessed LTL in 205 participants across glucose tolerance groups at baseline and after three years in the mixed ancestry population of South Africa which have been shown to have high rates of obesity and T2DM. Baseline and follow-up data included glucose tolerance status, anthropometric measurements, lipids, insulin, γ-glutamyl transferase (GGT), cotinine, and HbA1c. Telomere length was measured using the absolute telomere q-PCR method performed on a Bio-Rad MiniOpticon Detector. No significant difference was detected in LTL across glucose tolerance groups at both time points, including in subjects who showed a deterioration of their glucose tolerance status. There was, however, a significant negative correlation between LTL and age which was more pronounced in diabetes (*r* = −0.18, *p* = 0.04) and with GGT (*r* = −0.16, *p* = 0.027). This longitudinal study has demonstrated that LTL shortening is not evident within three years, nor is it associated with glycaemia. Further studies in a larger sample and over a longer time period is required to confirm these results.

## 1. Introduction

Telomeres are DNA-proteins situated at the ends of linear chromosomes and consist of non-coding, double-stranded, G-rich tandem repeats ranging in length from a few hundred base pairs in yeast to several kilo-base pairs in vertebrates [1]. The double stranded telomeric DNA is not all packaged into nucleosomes, and together with a single-stranded DNA overhang provides binding sites for a number of proteins [2]. In addition, it serves as a substrate for the enzyme telomerase which elongates and maintains telomere length [3]. The telomeric proteins play an important role in regulating telomere maintenance and any change in expression could result in accelerated shortening or dysfunction [4]. This multi-protein-DNA complex therefore forms a cap at the ends of chromosomes and is responsible for genome stabilization which is achieved by providing protection from degradation. It is thought that as cells replicate, telomeric DNA is sacrificed in order to preserve the coding regions of the genome [3,5]. This process continues until eventually a critical threshold is reached which results in cell senescence and/or apoptosis [1].

Telomere length and structure are regulated by complex mechanisms involving interactions between telomerase, the expression of associated proteins, such as TRF1 and TRF2 and telomere architecture [3,4]. It has, however, been observed that telomerase activity decreases with age and that telomere length is reduced in chronic inflammatory states. The mechanisms promoting this include increased levels of reactive oxygen species (ROS), decreased nitric oxide and decreased Telomerase reverse transcriptase (TERT) activity, all of which have been associated with the process of aging and atherogenesis [6]. Apart from the increased inflammatory activity and cell division, telomere erosion could also be caused by the release of ROS which causes damage to telomeric DNA by oxidative stress [7]. Both obesity and type 2 diabetes mellitus (T2DM) have been recognized as states of increased oxidative stress and it is understood that this could play a key role in progressive telomere shortening in these individuals [8].

T2DM is a metabolic disorder characterized by chronic inflammation, insulin resistance and increased risk of cardiovascular disease. It is the most common form of diabetes and accounts for 90–95% of all diabetes cases [9,10]. It is well recognized that obesity contributes to both the onset of hyperglycemia and the characteristic chronic pro-inflammatory environment. T-Lymphocytes and macrophages promote inflammation in adipose tissue by secreting pro-inflammatory cytokines, adipokines and chemokines, all of which have a direct effect on glucose intolerance and insulin resistance. The intensified inflammatory response promotes increased oxidative stress, elevated advanced glycation end-product (AGE) production and macrophage stimulation [11,12].

Numerous reports have observed significantly shorter leucocyte telomeres in patients with inflammatory diseases, however, evidence in T2DM remains scarce and controversial [9,13]. 

The mixed ancestry population of Cape Town is a heterogeneous South African ethnic group comprising 32–43% Khoisan, 20–36% Bantu-speaking African, 21–28% European and 9–11% Asian ancestry [14]. According to the 2011 population census of South Africa, this well described ethnic group [15] makes up 48% of the Western Cape population and 76% of the inhabitants of the Bellville South Community. [16]

Herein, we assessed leucocyte telomere length (LTL) in participants across all glucose tolerance groups at baseline and after three years in the mixed ancestry population of South Africa that have been shown to have high rates of obesity and T2DM [15]. 

## 2. Materials and Methods

### 2.1. Ethics

This investigation formed part of the Bellville South study which has been previously described [15]. Both the baseline and the three-year follow-up studies were approved by the Cape Peninsula University of Technology, Faculty of Health and Wellness Sciences Ethics Committee (Reference Number: CPUT/HW-REC 2008/002 and CPUT/HW-REC 2010 respectively). This sub-study was granted further ethical approval (CPUT/HW-REC 2017/H2) and was carried out according to the Code of Ethics of the World Medical Association (Declaration of Helsinki 2013). 

### 2.2. Study Design and Clinical Procedures

The study was longitudinal and involved participants of the well described Bellville South study [15,17]. Briefly, between 2008 and 2009 a cohort of mixed ancestry participants was recruited from the Bellville South region and three-year follow-up evaluation of the same participants commenced in February 2011 [17]. Participants were excluded if they were younger than 18 years, showed any clinical signs of infection, were pregnant or were using immunosuppressant drugs. Furthermore, participants were only included if they had fasted overnight and had not taken aspirin or anti-inflammatory drugs for a minimum of 14 days prior to sampling. Each individual filled in a standardized questionnaire which included questions regarding previous medical history, diet, as well as lifestyle habits, such as smoking and alcohol consumption.

All anthropometric measurements were performed according to World Health Organisation guidelines and performed in triplicate with the average used for the final analysis. Blood pressure measurements were taken using a semi-automated digital blood pressure monitor (Rossmax, Rossmax, San Diego, CA, USA) and weight was measured in kilograms (kg) using a calibrated Sunbeam EB710 digital bathroom scale (Sunbeam, Cape Town, South Africa) while participants were wearing light clothing and no shoes. Measurement of height was rounded off to the nearest centimeter (cm) and performed using a portable stadiometer. The body mass index (BMI) was then calculated by dividing the weight and height squared, BMI = $\frac{Body\;mass\;\left(\mathrm{Kg}\right)}{height\;{\left(m\right)}^{2}}$. 

A non-elastic tape was used to measure the waist circumference (WC) while subjects were standing in an erect position. Measurements were taken with the investigator in front of the participant, and by placing the measuring tape around the narrowest part of the torso as seen from the anterior view.

### 2.3. Blood Collection and Biochemical Assays

Following an overnight fast, blood samples were taken. In those participants with no history of diagnosed diabetes, a further sample was taken after an oral glucose tolerance test (OGTT). Samples for routine biochemical testing were sent to the Metropolis Private Pathology Laboratory, Cape Town which is an ISO 15,189 accredited medical laboratory. Blood glucose (mmol/L) was measured using the enzymatic hexokinase method while glycated hemoglobin (HbA1c) was determined by turbimetric inhibition assay (Cobas 6000, Roche Diagnostics; Mannheim, Germany). Insulin (mmol/L) was tested using a microparticle enzyme immunoassay (Axsym, Abbott, Abbott laboratories, Lake Bluff, IL, USA) and Ultra-sensitive (U-CRP) was analyzed by Latex Particle Immunoturbidimetric methods (Beckman AU (Beckman Coulter, Miami, FL, USA). Total cholesterol (TC), high density lipoprotein cholesterol (HDL-c), triglycerides (TG) and the liver enzyme γ-Glutamyltransferase (GGT) were analyzed using an enzymatic colorimetric method (Cobas 6000, Roche Diagnostics) while low density lipoprotein (LDL was calculated using Friedewald’s formula. The liver enzymes, alanine aminotransferase (AST) and aspartate transaminase (ALT) were measured using International Federation of Clinical Chemistry and Laboratory Medicine (IFCC) standardized reagents on a Beckman AU (Beckman Coulter, Miami, FL, USA) while serum cotinine was measured by Competitive Chemiluminescent on Immulite 2000 (Siemens, Berlin, Germany). Creatinine levels were determined on the Cobus 6000, Roche Diagnostics.

### 2.4. Glycaemic Status Classification

Classification of participants was done in accordance with the revised WHO criteria of 1999 [18]. History, fasting glucose, and 2-hour glucose following OGTT were used to classify participants as normotolerant, prediabetes (impaired fasting glycaemia, impaired glucose tolerance or combination of both), new diabetes, and known diabetes. 

### 2.5. DNA Extraction and Freeze Thawing Procedures

DNA was extracted using the salt extraction method (www.genomics.liv.ac.uk, 2001) from 1–2 mL of whole blood collected in EDTA. After extraction, the DNA was stored at −80 °C in a temperature-controlled freezer until analysis took place. On the day of analysis samples were taken out of storage and kept at 4 °C. The concentration and quality of the freeze-thawed DNA were both assessed using a nanodrop (Nanodrop Technologies, Wilmington, NC, USA) and samples included for analysis all had an OD (optical density) ratio ^A260^/_A280_ > 1.8.

### 2.6. Telomere Length Measurement

The method used to measure leucocyte telomere length was adopted from a protocol devised by O’Callaghan and Fenech [19]. Standards and primers used for the assay were obtained from Sigma-Aldrich/Merck (Saint Louie, MO, USA) and after purification and the removal of long oligomers (>50 mers), the oligomer standards and primers were diluted in Tris-EDTA to obtain a concentration of 100 µM. Thereafter they were vortexed, centrifuged for one minute and stored at −20 °C until required. To control for the plate effect, Standards and controls were placed on the first and the middle column and triplicates of samples were not place consecutively.

### 2.7. The Standard Curve for Human Single Copy Gene (SCG), 36B4 

PCR-grade water was used to dilute the telomere standard (84 mer oligonucleotide TTAGGG repeats). This was used to create a standard curve in order to calculate the number of repeats in each standard. A single copy gene (SCG) was included as a control and was used to determine the number of genome copies per sample. The 36B4 SCG is the most commonly used control, encodes the acidic ribosomal phosphoprotein (P0) and was used for all samples. 

### 2.8. Calculation of the Genome Copy Number

The synthesized 36B4 oligomer standard was 75bp in length with a molecular weight (MW) of 23,268.1. The weight of one molecule was MW/Avogadro’s number and therefore the weight of the standard was: 2.32681 × 10^4^/6.02 × 10^23^ = 0.38 × 10^−19^ g. The highest concentration standard, (36B4 SCG STD), had 200 pg of 36B4 oligomer (200 × 10^−12^ g) per reaction and hence contained: 200 × 10^−12^/0.38 × 10^−19^ = 5.26 × 10^9^ copies of 36B4 amplicon. The 36B4 SCG standard was equivalent to 2.63 × 10^9^ diploid genome copies, as there are two copies of 36B4 per diploid genome. A standard curve was produced by performing serial dilutions of the 36B4 SCG STD (10^−1^ to 10^−6^ dilution). Plasmid DNA (pBR322) was added to each standard to maintain a constant 20 ng of total DNA per reaction tube. The standard curve was used to measure the diploid genome copies per sample 

### 2.9. q-PCR for Absolute Telomere Length (aTL)

The SYBR Green KiCqStart ReadyMix (Sigma-Aldrich/Merck) was prepared as per the manufacturer’s specifications and after preparation, the master mix was mixed. Thereafter 16 µL was added into each of the 48 wells. In the wells reserved for the standards, 5.2 µL of diluted standard (10^−1^ through to 10^−6^) was added together with 2.8 µL of plasmid DNA (pBR322). 4 µL of the lung cancer cell line COLO 699 N (ECACC 93052608) positive control and 4 µL of nuclease-free water was aliquoted into a separate well as the negative template control (NTC). Lastly, 4 µL of the unknown DNA samples were aliquoted into the remaining reaction wells and the plate was sealed. The cycling conditions for both the telomere and the 36B4 amplicons were: 10 min at 95 °C, followed by 40 cycles of 95 °C for 15 s, 60 °C for 1 min, followed by a dissociation curve. 

Processing and analyzing of the data was done by exporting the values (Kb/reaction for telomere and genome copies/reaction for SCG) to csv format. The Kb/reaction value obtained was then used to determine the total telomere length in Kb per human diploid genome. This was done by dividing the telomere Kb/per reaction value by the diploid genome copy number to give a total telomeric length in Kb per human diploid genome.

### 2.10. Quality Control

After processing, the positive and no signal control (NTC) were checked together with the amplification in the standards and samples using results from the Telomere STD dilutions. Using these concentrations, the linear range of the reaction was observed. Those samples which fell outside this range were removed from further analysis. After amplification was completed the CFX Manager^TM^ Software (version 1.6.541.1028) produced a LOG starting value for each reaction that was equivalent to Kb/reaction based on the telomere standard curve values. To calculate the inter-assay and intra-assay coefficient of variations (CV) we performed 20 runs of the control 36B4 SCG in duplicate. The inter-assay CV was 4.8% and the intra-assay CV was 5.0% for the control sample. 

### 2.11. Statistical Analysis

Data were recorded on an Excel spreadsheet (Microsoft Office Professional 2010) and analyzed using Statistica (Dell^TM^ Statistica^TM^ 13.2, 2013) and the statistical software R (version 3.3.3, 2017-03-06), the R Foundation for statistical computing, Vienna, Austria). The Shapiro-Wilk W test was employed to determine whether continuous variables were normally distributed, based on probability thresholds of *p* > 0.1. Due to the departure from a normal distribution for many variables and failure of log transformation implemented in other studies to approximate normal distribution of TL values in our sample, general characteristics of the study participants are summarized as median (25th−75th), and comparisons across glucose tolerance status categories based on Kruskal-Wallis test. Partial correlations were used to investigate relationships between two continuous variables, while controlling for the potential effects of age and gender. Due to the lack of significant correlation between most variables and telomere length, further analyses via multivariable regressions were not pursued. A *p*-value of *p* < 0.05 was used to characterize statistically significant results. 

## 3. Results

### 3.1. General Characteristics of Participants at Baseline and Three Year Follow Up

Three hundred and fifty-one participants underwent both baseline and three-year follow-up evaluation. Of these 146 were excluded as they did not consent or did not have sufficient DNA, leaving 205 participants, median age 57 years and 78% being female. 

At baseline the glucose tolerance status of participants included 77 (37.6%) normoglycaemia, 43 (21%) pre-diabetes (IFG and/or IGT), 44 (21.4%) new diabetes and 41(20%) with known diabetes. As expected, the indices of glucose homeostasis, fasting blood glucose (FBG), post 2-hour blood glucose (PostBG) and glycated hemoglobin (HbA1c) were higher in subjects with prediabetes or diabetes (all *p* ≤ 0.0001) while there was no significant difference in Leucocyte Telomere Length at baseline and three-year follow-up across baseline glucose tolerance status (Table 1). 

During the three-year follow-up period 18 participants acquired a status of ‘progression’, including 10 who acquired a diabetes status. There was no significant difference in the LTL between those who had progressed and those who had remained stable and after three years, the length of the telomeres was similar to baseline levels (Figure 1 and Table 2).

### 3.2. Correlation between Telomere Length and Glucose Metabolic Parameters

Because no significant differences were observed across the glucose tolerance statuses and telomere length (Table 1), we recategorized participants into prediabetes and diabetes which includes both new diabetes and known diabetes. As expected there was a significant negative correlation between leucocyte telomere length and age at both baseline and three-year follow-up which was more pronounced in those subjects who had diabetes (*r* = −0.18, *p* = 0.04). Telomere length correlated negatively with GGT at baseline (*r* = −0.16, *p* = 0.027), and at 3-year follow-up there was a positive correlation with Insulin (*p* = 0.021) (Table 3). 

## 4. Discussion

The principal findings of this longitudinal study of mixed ancestry South Africans were that leucocyte telomere length (LTL) was not significantly different between the four glucose tolerance groups. Although high rates of deterioration of glucose status occurred over time there was no significant difference in the telomere length after three years, including in subjects who showed deterioration of their glucose tolerance status. There was a significant negative correlation between LTL and GGT at baseline which remained after adjustment for age and gender. 

Telomeres, the tandem repeats of TTAGGG DNA sequence situated at the ends of linear chromosomes shorten after each cell division [20]. As a result, the telomere length is an indicator of biological aging [21], as well as age-related disorders, such as type 2 diabetes [6,13,22]. In our study, we included participants with normal glucose tolerance, prediabetes, undiagnosed diabetes and those with diabetes on treatment. Telomere length was not significantly different across all these glucose tolerance status groups. 

Although a number of studies have shown an association between telomere length and diabetes [23,24,25,26,27,28,29,30,31] in some, this association was weak [28], cell type dependent [24] or associated with heart disease [30,31]. Sampson et al. measured the telomere length of peripheral blood monocytes and T-cells in Type 2 diabetes and normal controls. This study was performed using fluorescent in situ hybridization and significantly lower mean monocyte telomere length was reported in the diabetes group [24]. On the other hand, similar to our findings, some studies have failed to show an association between telomere length and diabetes. In one of these, Type 2 diabetes patients, without any complications demonstrated that, leucocyte telomere length was not significantly different to normoglycemic control subjects [30]. Similarly, in patients with type 2 diabetes, a median of 1 year from diagnosis, no difference in peripheral leucocyte TL was demonstrated when compared to age-matched non-diabetes control subjects [32]. Reasons for the variation in reported results could be attributed to a number of factors which include differing methodologies, patient populations, treatment and complications. 

Terminal restriction fragmentation has been described as the gold standard to measure telomere length, but due to the need for large quantities of DNA, a number of PCR based methods have been developed. These include q-PCR [33] and a modification of this technique, which incorporated the use of a standard curve to measure absolute telomere length [19]. 

Other researchers have utilized Quantitative Fluorescence in situ Hybridization (Q-FISH) and Flow –FISH, which has the advantage of sorting and isolating different haemopoietic cell populations using a flow cytometer before measuring telomere length using a peptide nucleic probe (PNA) [34]. 

Each of the above techniques has advantages and disadvantages which could lead to variations in the reported results. Causes of variation could include the quantity and quality of extracted DNA, variation in pipetting techniques, the cell population being analyzed and the expertise of the person performing the test [35]. In this study, the q-PCR method described by O’Callaghan et al. [19] was used and in order to overcome some of the challenges, the quality of the DNA was assessed using a nanodrop and only those with an OD (optical density) ratio ^A260^/_A280_ > 1.8 were used. Furthermore, strict quality control measures were adhered to and all samples were analyzed in triplicate. However, as the measurement of telomere length is not fully standardized, comparing research remains challenging and controversial. 

It is well recognized that telomere length shortens with age and occurs at a constant rate [21]. We however, did not find a significant difference in the telomere length after three years, including in subjects who progressed to a worse glucose tolerance status. It could be that a three-year timeframe is not a sufficient period of time or that the sample was too small to assess change in telomere length. 

In a study conducted on 2328 American Indian participants, LTL was assessed as a predictor of developing diabetes. Two hundred and ninety two developed the disease within a five year follow up period and it was determined that shorter LTL significantly and independently predicted an increased risk of developing diabetes [36]. In a similar study of 606 participants, 44 developed T2DM over a 15-year follow-up period and it was again concluded that there was a significant association between shorter leucocyte telomeres and risk of T2DM development [13]. In contrast however, an earlier study evaluating incident diabetes participants over a six-year follow-up timeframe could only demonstrate a weak association between LTL and diabetes risk and no statistically significant association was found [28].Therefore, while there is clear evidence for the effect of diabetes and its metabolic effects, such as low-grade inflammation and oxidative stress on telomere integrity, these contradictory reports are difficult to interpret and it appears that further large prospective studies over longer time frames are required. 

Hyperglycemia affects biochemical pathways leading to glucose oxidation, advanced glycation end product (AGE) formation and activation of polyol pathways which are all associated with the production of reactive oxygen species (ROS) and increased oxidative stress [37,38,39]. Studies have reported elevated levels of oxidative stress markers, such as glucose-6-phosphate dehydrogenase (G6PDH), malondialdehyde (MDA), glutathione (GSH), glutathione reductase (GR), glutathione peroxidase (GPx) and superoxide dismutase (SOD) [40] in Type 2 diabetes, while a negative correlation between oxidative stress and LTL has been demonstrated [26,41]. Further investigations in Type 1 and Type 2 diabetes examined oxidative stress by quantifying 8-hydroxy-desoxyguanosine (8-OHdG) and demonstrated a significant association with shortened telomeres in both types of diabetes [29]. A limitation of this current study was that we were unable to confirm this link as markers of oxidative stress were not included. 

Although telomeres are progressively lost with each cell division, a minimal extension does occur. This extension is attributed to the activity of telomerase, a reverse transcriptase enzyme [41]. Previous research has demonstrated a positive relationship between exercise and telomere length maintenance with subsequent lengthening. After a four-and-a-half-year follow-up period, an increase in LTL was observed in two-thirds of obese individuals with impaired glucose metabolism after significant lifestyle interventions were implemented. It was concluded from this that LTL can increase with time if positive healthy lifestyle interventions are initiated [41,42]. 

In our study, a significant negative correlation, which was more pronounced in subjects with diabetes, was observed between LTL and Gamma-glutamyl transferase (GGT). Gamma-glutamyl transferase is a sensitive but nonspecific index of liver dysfunction and could indicate excessive alcohol intake [43]. Previous studies have associated alcohol abuse with shortened telomeres, and premature cellular aging [44] which could provide an explanation for our findings. The link between high levels of GGT and the risk of developing Type 2 diabetes, however, remains unclear [45] but in a study of men aged 35-55 years, adjusted for confounders, such as BMI and alcohol drinking, it was found that the risk for developing diabetes increased with elevated GGT levels [46]. This study has been supported by previous work conducted among 1198 individuals from the Bellville-South region in Cape Town, in which GGT levels were independently associated with insulin sensitivity and obesity [47]. These studies suggest that chronic high levels of GGT may be an indicator of insulin resistance and this link may be important in the context of our results.

In summary, this longitudinal study has demonstrated no significant difference in the LTL between the baseline and the three year follow up. It is, however, important to highlight the limitations of this study which include the short follow up period, the small sample size and the mixed population which could have contributed to the increased variability in the telomere length. Furthermore, it is well documented that males have shorter telomeres than females [48] and as the majority of the participants were female (75%) this may have further influenced the results. A further limitation was that the inter-assay CV was below the recommended 7% even though the intra-assay CV was slightly higher. However, despite these limitations this study has provided important information and future work aims to include larger numbers of participants and over a longer period. 

Telomere length is influenced by a number of factors, including oxidative stress, high levels of inflammation and cell type, all of which were not fully explored in this study. Future investigations aim to examine the length of telomeres in different immune cell types and include oxidative stress markers, such as 8-OHdG. Telomere length has been well established as a marker of aging and age-related disorders and therefore additional knowledge of what influences telomere shortening in T2DM and other metabolic disorders could contribute to interventions that assist in the prevention of complications, such as cardiovascular disease.

## Figures and Tables

**Figure 1 cells-08-00464-f001:**
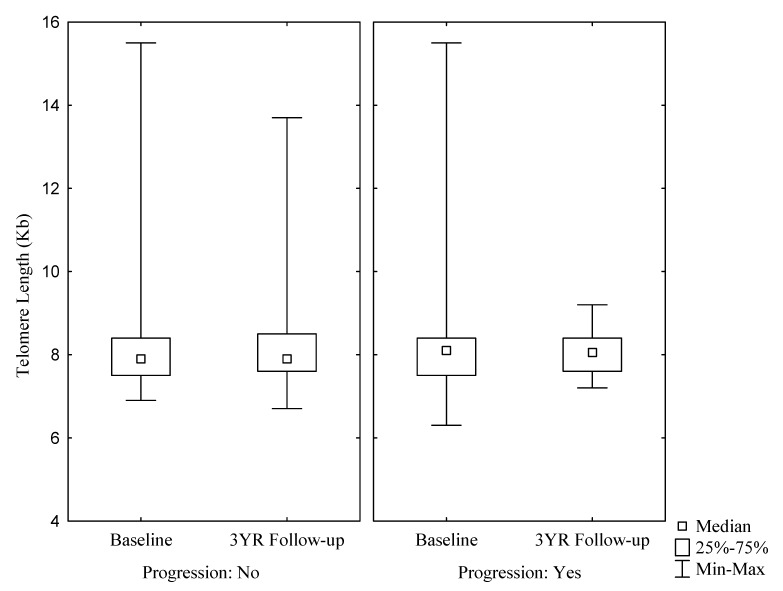
There were no significant differences between the telomere length at Baseline and 3-year Follow-up as categorized by progression and no-progression; Baseline and Follow-up for subjects who did not progress (N150): Median (range): 7.90 (7.50; 8.40) and 7.90 (7.60; 8.50); for subjects who did progress (N18): 8.10 (7.50; 8.40) and 8.05 (7.60; 8.40); *p* = 0.3328.

**Table 1 cells-08-00464-t001:** Differences in the biochemical and anthropometric measurements according to the glycemic profile: Baseline and three-year follow-up.

	Total N205	Normoglycemia N77	Pre-Diabetes N43	New Diabetes N44	Known Diabetes N41	*p*-Value	Total N205	Normoglycemia N91	Pre-Diabetes N37	New Diabetes N13	Known Diabetes N64	*p*-Value
	Baseline						Three-Year Follow-Up					
Age (Years)	57.0 (48.0; 62.0)	55.0 (46.0; 61.0)	55.0 (45.0; 62.0)	56.0 (47.5; 62.5)	59.0 (54.0; 64.0)	0.0869	59.0 (51.0;66.0)	58.0 (47.0;65.0)	57.0 (48.0;64.0)	60.0 (53.0;65.0)	61.5 (55.0;67.5)	0.0579
BMI	31.7 (27.4;36.5)	29.4 (25.0;33.4)	32.5 (28.7;36.3)	35.0 (30.5;38.1)	31.1 (26.5;37.3)	0.0002	31.3 (27.0;36.6)	29.4 (24.6;35.3)	34.5 (29.2;37.0)	32.9 (30.3;40.3)	31.9 (28.5;36.4)	0.0023
WC (cm)	101.5 (93.0;111.0)	96.0 (87.5;106.0)	105.0 (97.0;112.0)	107.7 (101.0;117.5)	101.0 (93.0;114.5)	0.0001	97.3 (88.3;107.0)	92.5 (81.1;105.0)	101.5 (92.3;108.0)	100.0 (99.3;117.3)	98.3 (91.5;109.0)	0.0047
HP (cm)	112.0 (104.0;122.7)	109.0 (103.0;116.0)	115.0 (108.0;122.7)	116.5 (108.7;127.0)	111.0 (99.8;125.5)	0.0036	108.8 (98.3;119.3)	105.1 (96.5;113.4)	114.7 (106.1;121.6)	110.7 (104.1;116.0)	110.0 (100.0;120.6)	0.0190
WHR	0.89 (0.85;0.95)	0.88 (0.84;0.92)	0.89 (0.83;0.95)	0.90 (0.86;0.98)	0.92 (0.86;0.96)	0.0371	0.88 (0.83;0.94)	0.88 (0.82;0.93)	0.87 (0.82;0.92)	0.94 (0.88;1.01)	0.91 (0.84;0.96)	0.0098
SBP (mmHg)	123.0 (112.0;135.0)	119.0 (111.0;129.0)	121.0 (112.0;132.0)	130.5 (116.0;140.5)	127.0 (114.0;142.0)	0.0259	136.0 (123.0;155.0)	131.0 (120.0;150.0)	142.0 (130.0;157.0)	130.0 (124.0;167.0)	139.0 (126.0;159.0)	0.0264
DBP (mmHg)	76.0 (68.0;83.0)	76.0 (67.0;85.0)	76.0 (68.0;82.0)	77.5 (69.0;86.0)	74.0 (68.0;83.0)	0.7670	82.0 (74.0;90.0)	80.0 (72.0;85.0)	87.0 (77.0;93.0)	84.0 (79.0;97.0)	81.0 (75.0;90.0)	0.0286
FBG (mmol/L)	6.00 (5.00;7.75)	5.10 (4.80;5.60)	6.00 (5.00;6.10)	7.90 (7.10;9.00)	11.05 (6.40;13.05)	<0.0001	5.60 (5.00;7.4)	5.00 (4.60;5.4)	5.80 (5.40;6.4)	8.90 (7.50;11.0)	8.55 (6.45;10.9)	<0.0001
PostBG (mmol/L)	7.40 (6.00;10.10)	6.00 (5.40;6.50)	8.60 (8.00;9.20)	13.00 (10.60;17.50)	NA	<0.0001	6.50 (5.30;8.10)	5.70 (4.90;6.60)	8.40 (7.90;9.00)	15.10 (11.80;19.00)	NA	<0.0001
Fasting Insulin (mIU/L)	9.20 (3.80;14.60)	8.30 (3.80;13.70)	10.20 (6.00;14.10)	10.70 (3.60;16.10)	7.80 (2.40; 14.15)	0.5562	12.20 (6.90;17.80)	10.40 (5.60;15.10)	13.20 (6.60;18.80)	17.90 (14.20;27.70)	13.20 (8.75;18.85)	0.0029
FBG/Insulin ratio	0.71 (0.43;1.62)	0.60 (0.37;1.24)	0.56 (0.40;0.95)	0.74 (0.54;2.22)	1.16 (0.74;3.16)	0.0029	0.56 (0.33;0.88)	0.48 (0.34;0.89)	0.46 (0.30;0.78)	0.50 (0.23;0.75)	0.64 (0.34;0.89)	0.5199
HbA1c (%)	6.10 (5.70;6.80)	5.70 (5.40;6.10)	5.90 (5.80;6.20)	6.75 (6.25;7.70)	7.35 (6.50;8.95)	<0.0001	6.20 (5.80;7.00)	5.90 (5.70;6.20)	6.20 (5.95;6.55)	7.40 (6.60;8.80)	7.20 (6.50;8.75)	<0.0001
U-CRP (mg/L)	5.50 (1.60;10.40)	4.00 (0.90;7.70)	7.60 (1.80;14.60)	7.10 (2.35;16.90)	4.10 (1.80;8.45)	0.0050	5.30 (2.00;9.20)	5.30 (1.80;11.80)	6.50 (3.90;10.10)	8.20 (4.40;10.40)	3.90 (1.40;7.00)	0.0874
GGT (IU/L)	30.0 (22.0;47.0)	27.0 (18.0;39.0)	31.0 (24.0;47.0)	39.0 (25.0;58.5)	29.5 (20.0;41.5)	0.0056	27.0 (20.0;45.0)	26.0 (19.0;45.0)	27.0 (20.0;46.0)	42.0 (36.0;47.0)	27.0 (20.0;41.5)	0.1010
TC (mmol/L)	5.56 (4.88;6.33)	5.52 (4.92;6.16)	5.53 (4.59;6.30)	5.82 (5.14;6.74)	5.43 (4.74;6.30)	0.3082	5.40 (4.71;6.21)	5.48 (4.76;6.29)	5.46 (5.00;6.08)	5.97 (4.85;6.13)	4.92 (4.53;6.36)	0.4994
TG (mmol/L)	1.37 (1.06;1.92)	1.23 (0.94;1.46)	1.44 (1.10;1.90)	1.58 (1.18;1.96)	1.70 (1.27;2.15)	0.0003	1.36 (0.99;1.75)	1.23 (0.88;1.69)	1.28 (1.01;1.68)	1.53 (1.36;1.94)	1.43 (1.13;1.95)	0.0099
HDL (mmol/L)	1.13 (0.99;1.39)	1.16 (1.00;1.42)	1.15 (1.06;1.39)	1.15 (1.00;1.36)	1.01 (0.86;1.23)	0.0212	1.33 (1.12;1.61)	1.39 (1.20;1.68)	1.21 (1.04;1.46)	1.34 (1.17;1.49)	1.25 (1.09;1.54)	0.0382
LDL (mmol/L)	3.65 (2.92;4.35)	3.63 (3.02;4.16)	3.63 (2.82;4.26)	3.88 (3.16;4.64)	3.38 (2.81;4.47)	0.3836	3.28 (2.68;4.10)	3.28 (2.73;4.03)	3.41 (3.03;4.21)	3.64 (2.75;4.32)	3.00 (2.38;4.06)	0.1795
TC/HDL ratio	4.78 (3.98;5.70)	4.52 (3.70;5.39)	4.51 (3.79;5.12)	4.89 (4.31;6.07)	5.05 (4.32;6.39)	0.0278	3.96 (3.13;4.98)	3.87 (2.97;4.73)	4.26 (3.27;5.52)	3.96 (3.40;5.19)	4.02 (3.19;5.01)	0.2066
S Cotinine (ng/mL)	9.00 (9.00;261.0)	10.00 (9.00;308.0)	9.00 (9.00;317.0)	9.00 (9.00;184.5)	9.00 (9.00;120.0)	0.0983	9.0 (9.0;289.0)	9.0 (9.0;339.0)	9.0 (9.0;163.0)	212.0 (9.0;334.0)	9.0 (9.0;40.6)	0.0665
Telomere length (kb)	7.80 (7.50; 8.40)	7.90 (7.50; 8.40)	7.80 (7.50; 8.30)	7.80 (7.50; 8.35)	8.10 (7.50; 8.60)	0.7618	7.90 (7.60; 8.50)	7.90 (7.60; 8.50)	7.80 (7.50; 8.40)	8.40 (8.00; 8.90)	7.90 (7.60; 8.40)	0.2204

Footnote: GGT, γ-Glutamyltransferase; BMI, Body mass index; SBP, Systolic blood pressure; DBP, Diastolic blood pressure; FBG, fasting blood glucose; PostBG, post 2-hour glucose; HP, Hip circumference; HDL, High density lipoprotein cholesterol; LDL, Low density lipoprotein cholesterol; TC, Total cholesterol; U-CRP, ultra-sensitive C-reactive protein; WC, Waist circumference; WHR, Waist to hip ratio; S cotinine, serum cotinine. *P*-values are from the Kruskal-Wallis test.

**Table 2 cells-08-00464-t002:** Telomere length as measured against the progression of the glucose tolerance status from baseline to three-year follow-up.

Glucose Tolerance Status	Telomere Length Baseline	Telomere Length Three-Year Follow-Up	
Baseline to Three-Year Follow-Up	Median (25,75Q)		*p*-Value
Normoglycemia to Prediabetes, *n* = 9	7.90 (7.50, 8.30)	7.90 (7.60, 8.20)	0.8385
Normoglycemia to Diabetes, *n* = 3	8.40 (7.40, 11.10)	8.40 (8.00, 8.60)	0.5930
Prediabetes to Diabetes, *n* = 6	8.05 (7.60, 14.80)	8.15 (7.60, 9.00)	0.7532

Footnote: *P*-values are from the Kruskal-Wallis test.

**Table 3 cells-08-00464-t003:** Partial correlation coefficients controlled for the effects for age and gender, for the association between telomere lengths and anthropometric and biochemical parameters according to glycemic status after the baseline and the three-year follow-up.

	Baseline						Three-Year Follow-Up					
Variable	Overall		Prediabetes		Diabetes		Overall		Prediabetes		Diabetes	
	*r*	*p*	*r*	*p*	*r*	*p*	*r*	*p*	*r*	*p*	*r*	*p*
BMI	−0.005	0.944	−0.007	0.937	0.134	0.397	0.044	0.538	0.123	0.198	−0.010	0.976
WC (cm)	−0.014	0.846	−0.001	0.993	0.238	0.130	0.022	0.753	0.112	0.243	−0.084	0.805
HP (cm)	0.012	0.868	−0.002	0.981	0.072	0.650	0.020	0.778	0.044	0.650	0.362	0.304
WHR	−0.048	0.499	−0.003	0.977	0.198	0.209	−0.003	0.968	0.095	0.330	−0.487	0.154
SBP (mmHg)	0.003	0.966	0.037	0.677	0.348	0.024	−0.065	0.358	0.020	0.836	−0.389	0.237
DBP (mmHg)	−0.021	0.763	−0.012	0.897	0.228	0.146	0.038	0.593	0.124	0.194	−0.154	0.650
FBG (mmol/L)	0.020	0.780	−0.007	0.937	−0.013	0.935	0.048	0.496	0.066	0.491	0.187	0.582
PostBG (mmol/L)	−0.023	0.772	−0.073	0.506	−0.151	0.339	0.146	0.091	0.216	0.149	−0.044	0.897
Fasting Insulin (mIU/L)	−0.017	0.811	0.000	0.997	0.079	0.625	0.022	0.755	0.146	0.125	0.681	0.021
FBG/Insulin ratio	−0.044	0.536	−0.040	0.662	−0.037	0.818	−0.010	0.888	−0.036	0.707	−0.123	0.719
HbA1c (%)	0.046	0.520	0.024	0.794	−0.022	0.888	0.050	0.478	0.083	0.385	0.150	0.659
U-CRP (mg/L)	0.073	0.300	0.116	0.198	−0.133	0.400	0.015	0.831	−0.030	0.756	−0.199	0.557
GGT (IU/L)	−0.156	0.027	−0.173	0.054	−0.176	0.264	0.127	0.072	0.092	0.335	0.096	0.778
TC (mmol/L)	0.024	0.733	0.035	0.699	0.168	0.289	−0.012	0.867	0.008	0.934	−0.224	0.508
TG (mmol/L)	−0.011	0.877	−0.058	0.520	−0.165	0.295	−0.002	0.978	0.007	0.938	−0.163	0.632
HDL (mmol/L)	0.109	0.121	0.073	0.416	0.193	0.220	0.041	0.558	0.017	0.856	0.171	0.615
LDL (mmol/L)	−0.003	0.969	0.043	0.637	0.224	0.153	−0.028	0.693	0.005	0.961	−0.223	0.510
TC/HDL ratio	−0.039	0.579	−0.001	0.989	−0.034	0.829	−0.040	0.573	−0.027	0.781	−0.337	0.310
S Cotinine (ng/mL)	0.065	0.357	0.077	0.396	−0.173	0.274	−0.050	0.482	−0.137	0.151	−0.512	0.107
S. Creatinine (umol/L)	0.058	0.408	0.100	0.269	0.138	0.382	−0.042	0.548	−0.041	0.665	0.317	0.342

Footnote: GGT, γ-Glutamyltransferase; BMI, Body mass index; SBP, Systolic blood pressure; DBP, Diastolic blood pressure; FBG, fasting blood glucose; PostBG, post 2-hour glucose; HP, Hip circumference; HDL, High density lipoprotein cholesterol; LDL, Low density lipoprotein cholesterol; TC, Total cholesterol; U-CRP, ultra-sensitive C-reactive protein; WC, Waist circumference; WHR, Waist to hip ratio; S cotinine, serum cotinine; S.Creatinine, serum creatinine. P-values are from Partial correlations.

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
