# Peer review of "Leucocyte Telomere Length and Glucose Tolerance Status in Mixed-Ancestry South Africans"

_cells, 2019, doi:10.3390/cells8050464_

Round 1

Reviewer 1 Report

-Recheck where to insert added sentences and paragraphs; added sentences appears out of context in several places.

-Box plots showed that telomere length measurement is possibly not normally distributed; if this is the case, a log transformation is needed. 

-Add a footnote to spell out abbreviations for the tables

-For BMI comparisons, are those median (range)?

Author Response

Point 1: Recheck where to insert added sentences and paragraphs; added sentences appears out of context in several places

Response 1: A thorough check of the manuscript has been performed and added sentences have been modified to ensure that the text flows correctly.

Point 2: Box plots showed that telomere length measurement is possibly not normally distributed; if this is the case, a log transformation is needed. 

Response 2: The reviewer is correct in saying that telomere length is not normally distributed and therefore medians were presented in Table 1 and for correlations, Spearmans correlation was used. The box plots only presents data on those who have progressed vs those who did not progress and medians were used. Log transformation would have been required if regression analysis was performed however this was not done in this case.

Point 3: Add a footnote to spell out abbreviations for the tables

Response 3: Thank you for highlighting this. All footnotes have been expanded to include all abbreviations.

Point 4: For BMI comparisons, are those median (range)?

Response 4: The answer to this question is yes. The statistics section has been amended to read: Data was presented as median and 25th -75th percentiles.

Reviewer 2 Report

Overall comments

The topic of the article is interesting but there some doubts about the technique used in TL measurement (samples are not run in triplicates as many articles suggest due to the variability when assessing TL), the design of the study especially since it is a heterogeneous population of 178 patients distributed in 4 groups, and also the statistical analysis performed.

Specific comments

1. There is no article from 2018 and 2019, the bibliography needs to be updated.

2. In the Methods section:

In the protocol you use they analyzed the samples in triplicate (“Individual samples are analysed in triplicate and accepted only if the standard deviation of the Ct values are <1Ct (CV >5%))” not in duplicates as mentioned in your article.

In the article it is written: “The inter-178 assay CV was 4.8 % and intra-assay CV was 5.0 % (bias 0.89%) for the control sample.” What are control samples?

In the protocol you use (The inter- and intra- experimental coefficient of variation of the 1301 telomere length measurement by absolute qPCR should be less than 7% and 2%, respectively) the CV is slightly higher than the values you obtained in 178 samples.

In the statistical analysis it is mentioned “In instances where the continuous variables were not normally distributed, Spearman correlation was used” What are the continuous variables that were not normally distributed?  

3.            In the Discussion Section:

There is a need to address the issue of the heterogeneous population of this article, a mixed ancestry population of South Africa. Besides TL depends on age, it is necessary to adjust each TL value for age of subjects. Smoking is an important factor that influence TL it is necessary to add information about the smoking status of patients.

Author Response

Point 1: The topic of the article is interesting but there some doubts about the technique used in TL measurement (samples are not run in triplicates as many articles suggest due to the variability when assessing TL), the design of the study especially since it is a heterogeneous population of 178 patients distributed in 4 groups, and also the statistical analysis performed.

Response 1: We acknowledge the confusion regarding whether samples were run in duplicate or triplicate. And yes indeed all samples were run in triplicate according to the method. The 20 runs of controls used to assess inter assay CV were run in duplicate. This has now been made clear within the article.

The heterogeneous or mixed ancestry population of the Bellville South community is well described and we have inserted a paragraph within the introduction to clarify this. This population makes up 78% of the Bellville South community and has been previously documented to have a high incidence of diabetes and obesity. 

Point 2:There is no article from 2018 and 2019, the bibliography needs to be updated.

Response 2: This point is acknowledged and an updated search of the literature has been conducted. Updated articles have been inserted into the reference section.

Point3: In the protocol you use they analyzed the samples in triplicate (“Individual samples are analysed in triplicate and accepted only if the standard deviation of the Ct values are <1Ct (CV >5%))” not in duplicates as mentioned in your article.

Response 3: We acknowledge that the manuscript was confusing on this point. Indeed samples were analysed in triplicate and the manuscript has been amended to explain this. (See response 1).

Point 4: In the article it is written: “The inter-178 assay CV was 4.8 % and intra-assay CV was 5.0 % (bias 0.89%) for the control sample.” What are control samples?

Response 4: This is an important observation and we thank the reviewer for raising this. In this study 36B4 SCG was used as a control and the difference in the inter assay and intra assay CV has been acknowledged and highlighted as a limitation.

Point 5: In the protocol you use (The inter- and intra- experimental coefficient of variation of the 1301 telomere length measurement by absolute qPCR should be less than 7% and 2%, respectively) the CV is slightly higher than the values you obtained in 178 samples.

Response 5 : This observation is correct and we have acknowledged this as a limitation. (See above).

Point 6:In the statistical analysis it is mentioned “In instances where the continuous variables were not normally distributed, Spearman correlation was used” What are the continuous variables that were not normally distributed?  

Response 6: We thank the reviewer for this point. In the end all our variables were not normally distributed and so we have clarified this by correcting the statistical section of the article.

Point 7: There is a need to address the issue of the heterogeneous population of this article, a mixed ancestry population of South Africa. Besides TL depends on age, it is necessary to adjust each TL value for age of subjects. Smoking is an important factor that influence TL it is necessary to add information about the smoking status of patients.

Response 7: The heterogeneous population studied in this article is well described and makes up 78% of the Bellville South Community according to Stats SA. This population group is known to have a high incidence of diabetes and obesity and has been previously studied. The introduction of the manuscript has been amended to include an explanation of this population group (see response 1 and amended manuscript.) Smoking and age does indeed influence telomere length and in order to analyse this cotinine levels were measured on all individuals. These results are included in table 1 and, table 2 shows that there was no correlation between cotinine levels and telomere length.

Reviewer 3 Report

In this manuscript, Weale and colleagues evaluated leukocyte telomere lengths (LTL) at baseline and after three years in 205 individuals with different glucose tolerance. The authors found no differences in telomere lengths among the groups at both baseline as well as after 3 years. The manuscript is well written but would be strengthened by considering the following comments.

Methods

Did authors investigate plate effects in telomere assay?

Results

In table 1, please use the same term for normoglycaemia.

There were individuals who newly became Normoglycaemic from pre-diabetic (based on Table 1). It would be interesting to see LTL changes in those individuals. Also, it would be more informative to state the disease status of 18 progressed participants (figure 1) and compare their LTL by the level of progress.

It is interesting that there was a negative correlation between LTL and GGT at baseline, the opposite was noted after 3 years of follow-up. In the discussion the authors described the association between GGT and insulin sensitivity. Have the authors examined the longitudinal ‘changes’ in GGT and LTL (not cross-sectional evaluation at baseline and at 3 year follow-up, which were presented in the manuscript) in the context of disease progression?

It is also interesting that there was a significant positive correlation between LTL and fasting insulin at 3-year follow-up among diabetic patients. Can the authors show the results stratified by disease progression status? It looks like the association is mainly driven by those who were newly diagnosed T2D (due to suboptimal control of disease), based on their fasting insulin level shown in Table 1.

Author Response

Comments and Suggestions for Authors

In this manuscript, Weale and colleagues evaluated leukocyte telomere lengths (LTL) at baseline and after three years in 205 individuals with different glucose tolerance. The authors found no differences in telomere lengths among the groups at both baseline as well as after 3 years. The manuscript is well written but would be strengthened by considering the following comments.

Methods

Did authors investigate plate effects in telomere assay?

Response: To control for the plate effect, Standards and controls were placed on the first and the middle column and triplicates of samples were not place consecutively.

Results

In table 1, please use the same term for normoglycaemia.

Response: Thank you for raising this point. We have changed accordingly

There were individuals who newly became Normoglycaemic from pre-diabetic (based on Table 1). It would be interesting to see LTL changes in those individuals. Also, it would be more informative to state the disease status of 18 progressed participants (figure 1) and compare their LTL by the level of progress.

Response: Figure 1 depicts changes in TL in those 18 individuals who progressed. We have also added a table of the disease status of the 18 individuals.

It is interesting that there was a negative correlation between LTL and GGT at baseline, the opposite was noted after 3 years of follow-up. In the discussion the authors described the association between GGT and insulin sensitivity. Have the authors examined the longitudinal ‘changes’ in GGT and LTL (not cross-sectional evaluation at baseline and at 3 year follow-up, which were presented in the manuscript) in the context of disease progression?

It is also interesting that there was a significant positive correlation between LTL and fasting insulin at 3-year follow-up among diabetic patients. Can the authors show the results stratified by disease progression status? It looks like the association is mainly driven by those who were newly diagnosed T2D (due to suboptimal control of disease), based on their fasting insulin level shown in Table 1.

Response: Due to the small numbers (18) of progressed individuals, it would not be statistically meaningful to conduct this. Please see new Table 2.

Round 2

Reviewer 2 Report

I stated that the manuscript needs a major revision…and they have changed several lines alog the text.  

As indicated in the first review the statistical analysis performed is very poor and needs a big improvement. They need the assistance of a statistician. It is needed to redo all the statistical analyses for instances they should perform ANOVA including in the different models the confusion variables such as age, sex, ethnicity, smoking and others.  

Author Response

Point 1:

I stated that the manuscript needs a major revision…and they have changed several lines alog the text.  

As indicated in the first review the statistical analysis performed is very poor and needs a big improvement. They need the assistance of a statistician. It is needed to redo all the statistical analyses for instances they should perform ANOVA including in the different models the confusion variables such as age, sex, ethnicity, smoking and others.

Response:

We thank the reviewer for this additional comment. We however believe that some of the queries raised by the reviewer are due to the lack of sufficient clarity in our initial description of the statistical analyses performed. We have therefore done further edition of the stat analyses section, heading and footnotes of table to bring more clarity. As already stated in the manuscript, we formally tested the normal distribution of continuous variables in our sample. Because TL and many other variables did not follow a normal distribution either in the overall sample or in glucose tolerance status categories; we used the Kruskal-Wallis test (which is a non-parametric equivalent of the ANOVA test the reviewer is referring to), to compare participants’ characteristics across glucose tolerance status subgroup. We believe the non-parametric tests are more appropriate than ANOVA when the assumption of normal distribution is violated, unless appropriate variable transformation functions are identified and applied to approximate normal distribution before implementing ANOVA procedures. The further emphasized the need of conducting multivariable regression models to account for possible effect of confounding factor. We also want to stress that coefficients in our table 2 are partial correlation coefficients that account for the possible effect of age and gender. With this very basic adjustment, we are seeing virtually no association between all the attribute tested and TL either at baseline and at 3 years. Had we seen an association between few variables and TL, then it would motivate conducting multivariable regressions to mutually adjust for the effect of significant predictors in the basic model. In the absence of association after basic adjustment, there is really no ground to pursue multivariable regressions, which obviously will confirm the lack of association. In the presence of our small sample (particularly in some glucose tolerance subgroups) we controlled only for the effect of age and sex to preserve our statistical power; the rational for choosing age and gender being that most variables (measured or unmeasured) would tend to correlate with age and gender. It is of note that this work was conducted in a mono-ethnic sample, and therefore no room for adjustment for ethnicity.

Round 3

Reviewer 2 Report

The authors should include “baseline and three year follow-up evaluation” in the Legend of Tables 1 and 2. An also they should indicate the corresponding statistical analysis in the Footnote of Table 2.

The authors are right, telomere length measurement does not follow a normal distribution but in many papers, data are transformed to better fit a normal distribution (for example: log transformed TL). This is more appropriate than working with crude data.

The limitations of the article should be listed more clearly in the Discussion section: the mixed population, age and sex differences, the small sample size, the high variability of TL and so on.

Author Response

Comments and Suggestions for Authors

The authors should include “baseline and three-year follow-up evaluation” in the Legend of Tables 1 and 2. An also they should indicate the corresponding statistical analysis in the Footnote of Table 2.

Response: Corrected accordingly

The authors are right, telomere length measurement does not follow a normal distribution but in many papers, data are transformed to better fit a normal distribution (for example: log transformed TL). This is more appropriate than working with crude data.

Response: After log transformation of the data, a normal distribution was not achieved. Therefore, we could not use log transformed data as used in other studies. The next best thing was to use the median which is recommended in such cases.

The limitations of the article should be listed more clearly in the Discussion section: the mixed population, age and sex differences, the small sample size, the high variability of TL and so on.

Response: Thank you for raising this point. We have changed the limitations accordingly.